# Switchable Piezoresistive SmS Thin Films on Large Area

**DOI:** 10.3390/s19204390

**Published:** 2019-10-11

**Authors:** Andreas Sousanis, Dirk Poelman, Christophe Detavernier, Philippe F. Smet

**Affiliations:** 1Lumilab, Department of Solid State Sciences, Ghent University, Krijgslaan 281/S1, 9000 Ghent, Belgium; andreas.sousanis@ugent.be (A.S.); dirk.poelman@ugent.be (D.P.); 2Cocoon, Department of Solid State Sciences, Ghent University, Krijgslaan 281/S1, 9000 Ghent, Belgium; Christophe.Detavernier@UGent.be

**Keywords:** SmS, switchable materials, semiconductor to metal transition (SMT), optical properties, structural properties

## Abstract

Samarium monosulfide (SmS) is a switchable material, showing a pressure-induced semiconductor to metal transition. As such, it can be used in different applications such as piezoresistive sensors and memory devices. In this work, we present how e-beam sublimation of samarium metal in a reactive atmosphere can be used for the deposition of semiconducting SmS thin films on 150 mm diameter silicon wafers. The deposition parameters influencing the composition and properties of the thin films are evaluated, such as the deposition rate of Sm metal, the substrate temperature and the H_2_S partial pressure. We then present the changes in the optical, structural and electrical properties of this compound after the pressure-induced switching to the metallic state. The back-switching and stability of SmS thin films are studied as a function of temperature and atmosphere via in-situ X-ray diffraction. The thermally induced back switching initiates at 250 °C, while above 500 °C, Sm_2_O_2_S is formed. Lastly, we explore the possibility to determine the valence state of the samarium ions by means of X-ray photoelectron spectroscopy.

## 1. Introduction

Samarium monosulfide (SmS) is a chalcogenide material which presents a pressure-induced transition from the semiconducting (also called “blue or black state”) to the metallic (or golden) state [1,2,3,4]. At room temperature, this transition takes place at a pressure of 0.65 GPa for bulk crystals [5] and is accompanied by strong changes in the optical and electrical properties. The transition can also be optically-induced, positioning SmS among other types of switchable materials, such as VO_2_ and GeSbTe-based materials [6,7,8,9,10,11,12]. While bulk metallic SmS returns to the semiconducting phase upon release of the applied pressure, this is not necessarily the case for SmS thin films. The semiconducting state can however be restored by thermal annealing [13,14,15]. Although the transition from semiconducting to metallic state is abrupt, being accompanied by large changes in both electrical and optical properties, the only structural change is a volume collapse of the SmS unit cell, without any phase transition [16]. The applied pressure increases the crystal field splitting of the 5d state of the Sm^2+^ ion, promoting the bottom of the 5*d*(t_2g_) band to touch the (occupied) 4f^6^ states of the Sm ion. After that, the Sm ion, initially in the divalent state, promotes an electron to the (5d) conduction band, and it finally becomes Sm^3+^ [17]. The associated volume collapse (up to 15%, [18]) is related to the much smaller size of the Sm^3+^ ion with respect to the Sm^2+^ ion. SmS also features an additional gradual transition from the paramagnetic state to an anti-ferromagnetically ordered state at about 1.85 GPa ([19,20] and references therein). Upon this transition, a lot of interesting phenomena emerge, such as semi-metal behavior and Kondo-like insulating properties [21]. During the last years, several techniques have been used to grow SmS thin films, such as pulsed laser deposition (PLD) [22], molecular organometallic chemical vapor deposition (MOCVD) [23], magnetron sputtering in both r.f. and d.c. operation [24], and electron beam evaporation [17]. A detailed presentation of the properties of SmS can be found in our recent review paper [4]. In the present work, we explore the experimental conditions of the deposition of SmS thin films on large areas, as accurate control over deposition temperature and chemical composition is required to arrive at the desired stoichiometry [25]. After determining the optimized deposition conditions, we report on the optical, structural, and piezoresistive properties, before, after, and during the pressure-induced switching.

## 2. Materials and Methods

SmS thin films were grown by reactive e-beam sublimation of Sm metal (Smart Elements, 99.99%) in an H_2_S atmosphere (Praxair, 99.8%) on Corning glass substrates (1737F) and Si (100) wafers covered with native oxide (~2 nm). The base pressure for the depositions was typically 2 × 10^−6^ mbar (Leybold Univex 450 deposition system). Deposition rates (varying from 0.8 to 1.1 nm/s) were monitored with the help of a pre-calibrated non-heated quartz balance crystal. In initial experiments, the H_2_S flow was controlled manually using a needle valve and the amount of H_2_S introduced in the chamber was tuned to a partial pressure of 8 × 10^−6^ mbar (with a corresponding substrate temperature of 230 °C). In all subsequent experiments, including the optimization of the deposition parameters, an H_2_S flow controller (Bronkhorst, EL-FLOW Select) was used. In that case, the H_2_S flow was tuned in order to achieve a specific partial H_2_S pressure in the chamber. Halogen lamps were used to indirectly heat the substrate via a 150 mm (6 inch) diameter substrate holder made of copper.

Thin films were rubbed with a blunt, smooth metallic rod, without visibly damaging the films, to induce the metallic state of SmS (M-SmS). To switch the films back to the semiconducting state, different thermal annealing methods were explored. Although annealing at 400 °C in vacuum is an easy way to switch the M-SmS back to S-SmS, the time-consuming pumping, heating and cooling rates limit the swiftness of the experimental process. Using an in-situ XRD setup, we were also able to monitor the thermally-induced (rate 10 °C/s) back switching, faster than annealing in vacuum, maintaining the ability to choose specific atmospheres.

X-ray diffraction (XRD) was measured using a standard diffractometer (Siemens D8 with Ni-filtered CuK$\alpha_{1}$ radiation, λ = 0.154 nm). Selected samples were viewed via scanning electron microscopy (SEM). SEM analysis was carried out in a FEI Quanta FEG 200 electron microscope, with a point resolution of 1.7 nm at 20 kV.

In situ high temperature X-ray diffraction (HTXRD) patterns were obtained using a Bruker D8 Discover XRD system with an integrated annealing chamber in order to evaluate the thermal stability of the thin films [26]. The XRD system was equipped with a linear X-ray detector and a copper X-ray source (λ = 0.154 nm). By using a linear detector, the source and the detector could be kept at fixed positions, while snapshots of the diffracted X-rays were collected every few seconds within a 2θ window of approximately 17 degrees. The system was first pumped down to a pressure below 5 × 10^−2^ mbar and was then filled with the appropriate gas (O_2_, He or combination of O_2_ and He) with a total pressure in the chamber slightly above atmospheric pressure.

The specular transmission and reflectance spectra were recorded at room temperature with a Varian Cary 500 UV–vis spectrophotometer in the wavelength range 200–800 nm.

Piezoresistance measurements were carried out using a homemade setup. The main principle is based on a conventional 4-point probe setup (supported by a Keithley 2010 multi-meter), which can measure the resistance of a sample as a function of the force applied by a spherical indenter [27]. Two wires of the 4-point probe setup are connected to the indenter (CuBe with radius of 0.685 mm), while two additional copper wires are directly connected to two other spots of the sample, which are not pressurized by the indenter. In this way, the measurement yields the contact resistance as a function of applied force. When using top and bottom electrodes, the resistance across the thin film can also be measured. Bottom and top electrodes consisted of a 50 nm thick Al thin film, deposited by e-beam evaporation.

The used X-ray photoelectron spectroscopy (XPS) set-up was an ESCA S-probe VG. The X-ray source used an Al (Ka)/(1486.6 eV) anode and the base pressure was around 5 × 10^−10^ mbar. To study the chemical properties across the SmS layer Ar^+^ was used to sputter away the material layer by layer. When Ar^+^ was used for depth profiling analysis, the pressure of the system reached pressures up to 2 × 10^−7^ mbar.

## 3. Results and Discussion

### 3.1. Basic Properties of SmS Thin Films

E-beam evaporation starting from a SmS target can be used to deposit SmS thin films. However, the direct synthesis of SmS in powder form is not straightforward, as under sulfurizing conditions Sm_2_S_3_ is easily formed. Consequently, other research groups have employed co-deposition of Sm_2_S_3_ and Sm metal to arrive at the desired stoichiometry [28]. In this work [13], Sm metal was deposited in a reactive H_2_S atmosphere, in order to obtain the formation of SmS at the substrate. Both fluxes (Sm, S) need to be carefully balanced. When the sulfur flux is too low, Sm metal can be deposited, while a too high flux leads to the formation of sulfur rich compounds, like Sm_3_S_4_ or Sm_2_S_3_. Figure 1 shows the essential features of the switching ability (Figure 1a) of a SmS thin film deposited at 250 °C, with H_2_S pressure of 1 × 10^−5^ mbar and a 0.8 nm/s deposition rate of the Sm metal. The thin film is polycrystalline (Figure 1b,c) and the diffraction peaks confirm that the as-deposited thin film is in the semiconducting state (S-SmS), corresponding to a high resistive state at ambient pressure (Figure 1d). The lattice constant, as calculated from XRD, is 587 (±1) pm. This is smaller than reported values for bulk SmS crystals (597 pm, [29]), but comparable to other reports on thin films [22]. When applying pressure, the resistance across the thin film strongly drops when the applied force exceeds 0.2 N, and a low resistive state is obtained (measured on a thin film stack with bottom and top electrodes, where the SmS thin film was deposited under similar conditions as for the single SmS thin films). Upon release of the pressure, the low resistive state is maintained and the thin film has a metallic appearance (M-SmS). The X-ray diffraction pattern (Figure 1b) confirms this, as the (111) and (200) diffraction peaks are shifted to higher diffraction angles, compatible with the expected reduction in unit cell parameters. The calculated lattice constant in this state is 576 (±2) pm, which is larger than in the case of metallic bulk crystals (562 pm), indicating that the metallic state contains a mixture of Sm^2+^ and Sm^3+^ ions [30]. Literature values for thin films are typically around 570 pm ([13] and refs. therein). The fact that the M-SmS state can be stabilized without pressure is a different behavior in SmS thin films compared to bulk crystals, as the latter switch back to the semiconducting state upon pressure release [31]. In thin films, the substrate apparently plays an important role as it can provide additional stress affecting the needed pressure for the (back)switching, similar to the effect of clamping in the case of piezo-electric thin films [4,28]. Nevertheless, thermal annealing can be used to convert a M-SmS thin film back into S-SmS [13]. Figure 1c shows the evolution of the position of the (200) diffraction peak as a function of annealing temperature (measured in the HTXRD setup, using an oxygen atmosphere). Up to 200 °C, the M-SmS state is stable. While the diffraction peaks start to shift at about 250 °C, only at 400 °C a full switch back to the diffraction angles for S-SmS is obtained. In Figure 1c, the black dot stands for the (200) peak position of the as-deposited S-SmS, prior to the induced transition to the metallic state. This procedure—application of pressure and thermal switch back—can be repeated multiple times, as is shown in the optical behavior of the thin films. Figure 1a shows the absorbance measurements for a 100 nm SmS thin film deposited on glass, initially in the semiconducting state. After application of pressure, by rubbing the surface, the M-SmS state is obtained, which leads to a stronger reflection in the visible part of the spectrum, lowering the transmission through the film [17]. The sample was switched back to the semiconducting state by thermal annealing in vacuum. We repeated this experiment several times, confirming the repeatable switching character of SmS thin films. Previous investigations have shown that a suitable temperature for switching thin films under vacuum annealing is at about 400 °C [32]. We used one hour at 400 °C under vacuum for switching back. At shorter annealing times (40 min) an incomplete switching back was observed, while at longer annealing times (or higher temperatures) degradation could be seen. This degradation could be related to the oxidation of a small part of the thin film surface, even under vacuum conditions. The chemical stability of the thin films under different thermal annealing conditions will be discussed in detail in Section 3.4.

### 3.2. Optimizing Synthesis Conditions

The thin films discussed in the previous section had been deposited using optimized conditions for the Sm deposition rate and the H_2_S flux. In this section we briefly discuss this optimization process. In principle, only the ratio between both fluxes determines the eventual stoichiometry, not the absolute fluxes. However, at low deposition rates, the probability for the reaction of the Sm atoms arriving at the substrate with trace gases increases. Hence, we aimed for a relatively fast Sm evaporation rate of around 0.8 to 1 nm/s. The matching H_2_S flux is expressed as the partial pressure in the deposition chamber. It is important to note that H_2_S gas was delivered by a conical shape tube, positioned sufficiently far from the substrate, in order for the opening angle of the gas flux to cover the entire area of the substrate. As such, the deposition conditions are tool and configuration dependent, requiring to be re-established when, e.g., the geometry or target-to-substrate distance is changed.

Figure 2 demonstrates the significant influence of relatively small changes in deposition parameters on the structural properties of SmS thin films. For a predefined substrate temperature of 250 °C, we evaluated different Sm deposition rates (amount of Sm metal) and H_2_S pressure values. Using a deposition rate of 1.0 nm/s and a partial pressure of H_2_S 1 × 10^−5^ mbar, a SmS thin film with broad XRD peaks (sample a) was obtained. At a partial pressure of H_2_S 9 × 10^−6^ mbar and 0.9 nm/s Sm deposition rate (sample b, which is a similar Sm/H_2_S ratio as in sample a), a similar diffraction pattern is found. Slightly lowering the Sm:S ratio (deposition rate of 0.8 nm/s and H_2_S pressure of 9 × 10^−6^ mbar (sample c) to 1 × 10^−5^ mbar (sample d)) provides high-quality, switchable S-SmS thin films. The reduction in XRD peak width (from sample a to d) also likely demonstrates that the grain size increases, although it could also be related to an improved stoichiometry of the thin film, being close to SmS. Further lowering the Sm:S ratio, radically changes the structural properties of the thin films. Indeed, in sample e we observe an intense peak at around 29°, which corresponds to Sm_3_S_4_, a compound with S excess [33].

After finding the optimum partial pressure (between 9 × 10^−6^–1 × 10^−5^ mbar for a deposition rate of 0.8 nm/s), corresponding to the correct amount of H_2_S flow, we checked the influence of the substrate temperature on the crystallization of the thin films. The temperature does not play a significant role within the tested range from 150 to 300 °C in terms of the formation of (semiconducting) SmS itself. All thin films can be switched to the metallic state by rubbing (Figure 3a). The position of the diffraction peaks (and thus also the lattice constant) hardly changes for both states, as a function of substrate temperature. Nevertheless, the substrate temperature has some impact on the thin film crystallization, as deposition at 150 °C leads to somewhat broader and weaker diffraction peaks (Figure 3b). This could be related to smaller crystallite sizes or small variations in terms of the Sm:S stoichiometry. A substrate temperature of 250 °C appears to be an optimum value, yielding the lowest FWHM values for the diffraction peaks, while showing good switching ability and crystallization.

Sheet resistance ($R_{s}$) measurements were performed using a conventional four-point probe set-up for samples deposited at different temperatures. For this purpose, we initially deposited a 600 nm thick insulating Al_2_O_3_ on Si (100), before the SmS thin films were deposited on top. The measured sheet resistance values (Table 1) were converted into resistivity values ($R_{s}=\rho /t$, with t the thickness of 100 nm). The as-deposited S-SmS thin films showed values between 1.5 × 10^−1^
Ωcm and 5 × 10^−1^
Ωcm, while the corresponding values of the scratched M-SmS thin films presented values from 3 × 10^−3^
Ωcm to 5 × 10^−3^
Ωcm, with no clear dependency on the deposition temperature. Literature values for the semiconducting and metallic state of thin films are 1.4 Ωcm and 1.1 × 10^−3^
Ωcm, respectively [22]. In case of bulk crystals, the resistivity in the semiconducting and the metallic state were found to be at around 4 × 10^−1^
Ωcm and 4 × 10^−3^
Ωcm, respectively [30]. This significant resistivity change between both states can be very useful for sensing applications and nano-electronics. Hence, ways to induce a full reversibility between the two states of SmS thin films, without thermal annealing treatment or tensile force, is of great importance for oncoming research.

### 3.3. Homogeneous, Switchable SmS Thin Films on 150 mm Diameter Wafers

For applications in the context of piezoelectronic devices [34], it is important to achieve a homogeneous deposition on wafer scale sizes. For those purposes, a proper substrate holder and heater was constructed, which can support deposition onto wafers with a diameter of 150 mm (or 6 inch).

A first step to evaluate the homogeneity of the as-deposited thin films on 150 mm diameter wafers consisted of measuring the optical reflectance of the sample at several different points across the wafer area (Figure 4). All of the curves are quite similar, typical of the S-SmS state. The small differences are presumably due to thin film interference effects, following slight changes in thickness across the wafer surface. Nevertheless, the deposited S-SmS thin film looks visually homogeneous over the entire surface (as illustrated in the bottom photograph in Figure 4, for a piece of about 3 cm by 3 cm). It was verified that the metallic state could be obtained by rubbing anywhere on the sample. One representative reflection spectrum (red curve) for the M-SmS state is displayed in Figure 4, as well.

In order to evaluate the structural properties, XRD measurements were performed on 18 different spots evenly spread across the surface of the wafer (Figure 5). The spot size of the X-ray source was 2 mm in diameter. The selected measured spots are a good approximation for the homogeneity across the wafer area. The entire wafer surface was covered by as-deposited S-SmS. The intensity and FWHM of the (200) diffraction peak can be taken as a figure of merit for the deposition homogeneity, as the ratio between the intensities for the (111) and (200) diffraction peaks was fairly constant over the entire wafer area (not shown), pointing at a similar texturing. The intensity values were obtained by subtracting the background of each XRD measurement. Similar FWHM values are obtained over the entire surface, while the diffraction intensity reveals some areas near the edges showing slightly lower values (Figure 5). The different behavior at the edges of the wafer can be explained by the spatial distribution of the Sm and H_2_S fluxes, with the latter one probably the least controlled in the present setup. Further optimization of the H_2_S gas inlet method would presumably further homogenize the composition of the thin film. Nevertheless, this is the first time to the best of our knowledge that a switchable SmS thin film is deposited on such a large scale. Recently, co-sputtering was reported to be able to handle the deposition of another piezoresistive Sm-chalcogenide, SmSe, on substrates with a diameter of 200 mm [34].

### 3.4. Thermal Stability of SmS

In-situ XRD measurements of SmS thin films were performed, under three different atmospheres, up to 800 °C (rate: 10 °C/s) to evaluate their chemical stability (Figure 6). Figure 6b shows the behavior of the as-deposited S-SmS film under 20% O_2_ and 80% He (ambient-like) atmosphere. The (200) diffraction peak of S-SmS is dominant up to 500 °C. At 535 °C, the emerging peak at 2θ = 29.2° is related to Sm_2_O_2_S, while two more peaks related to Sm_2_O_2_S appeared at the same temperature, at around 26° and 37.5°. When the atmosphere is 100% O_2_ (see Figure 6a), we observe a similar behavior. Using an oxygen free atmosphere (100% He, Figure 6c), the occurrence of Sm_3_S_4_ (diffraction peak at 29.5°) is observed in the temperature interval from 380 °C to 535 °C, before Sm_2_O_2_S becomes dominant at higher temperatures [33,35].

Similar experiments were performed for the M-SmS thin films (Figure 6d–f). In all cases, the (200) peak shifts to lower 2θ values, corresponding to the acquisition of the semiconducting state (see, Figure 1c for a more quantitative presentation of the back switching). This transition is accompanied by the acquisition of larger lattice parameter as the radius of Sm^2+^ is bigger than that of Sm^3+^ ions. Also, both Sm_2_O_2_S and Sm_3_S_4_ appear at the same temperatures, as in the case of S-SmS, showing similar chemical stability of both states of SmS. Lastly, we also conducted annealing in air, which demonstrated that SmS layers in both states react with oxygen when the temperature reaches 400 °C.

### 3.5. Using XPS to Probe the Valence State of Sm

X-ray photoelectron spectroscopy (XPS) is in principle a useful tool to study chemical properties of materials and the valence state and bonds of the constituent elements. Our intention was to use XPS analysis to evaluate the valence state of the Sm ions in S-SmS thin films and to study the extent of thin film oxidation. The Sm 3d_5/2_ photoelectron peak showed the presence of both Sm^2+^ and Sm^3+^ ions for an as-deposited SmS thin film (Figure 7a). On the one hand this is surprising, as the optical and electrical characterization showed the presence of SmS in the semiconducting state. On the other hand, the XRD patterns indicated that the lattice constant of those S-SmS thin films is smaller than the lattice constant for bulk SmS crystals, pointing at a mixed Sm^2+^–Sm^3+^ composition, as noticed earlier. Unfortunately, increasing the XPS measurement time in order to record more detailed XPS spectra showed that the shape of the photoelectron peak was not stable (Figure 7a), with longer exposure times to the X-rays increasing the Sm^3+^/Sm^2+^ ratio.

Several parameters can drastically influence the measured ratio between the Sm valence states: (1) changes of the Sm oxidation state at the sample surface as a function of time due to dangling bonds, surface reconstruction or surface oxidation, even in ultrahigh vacuum, (2) an effect of the XPS X-ray beam itself on the Sm oxidation state and (3) changes in oxidation state induced by Ar-ion sputtering when depth profiling in XPS. Hence dedicated experiments were performed in order to determine the main parameters. Prior to the experiments shown in Figure 7a–c, the surface of the studied samples was sputtered away for 150 s, which corresponds to a sputter depth of about 50 nm. This etching process is meant to provide a better look at the Sm valence state, limiting any influence from possible surface oxidation (likely increasing the Sm^3+^ contribution), when the layer is stored in ambient conditions.

Figure 7a shows the strong change in the Sm valence state (Sm 3d_5/2_ transition), for a 100 nm as-deposited S-SmS (after 150 s of Ar-ion sputtering), under ultrahigh vacuum (UHV) conditions. At relatively short exposure time of 5 min after the end of the etching process (red curve), we see that Sm^2+^ dominates over Sm^3+^, while after 172.8 min, Sm^3+^ becomes clearly dominant. The full profile of the evolution of the Sm valence states, for an as-deposited S-SmS, can be seen in Figure 7b. During those measurements, the X-rays were incident on the sample during the entire measurement procedure. In order to check whether the X-rays were responsible for any of the conversion of the Sm oxidation state, the same type of measurement was performed, but with the X-rays switched off between two distinct measurements (shown by the stars in Figure 7b). The percentages for the two states (Sm^3+^ and Sm^2+^) were determined immediately after the initial sputtering cycle of 150 s, and then again after a waiting time of 60 min with the X-ray beam switched off. In this case, the Sm^3+^ fraction had also increased, from 60% to 75%, which is similar than for the case when the X-rays remained on. Hence, even under the UHV conditions, part of the measured Sm^2+^ ions convert to a Sm^3+^ state when the surface remains exposed. Our next step was related to possible changes in oxygen concentration that would potentially lead to Sm oxides and thus an increase of Sm^3+^. Indeed, the recorded spectrum of the O 1s photoelectron peak (at ~529–530 eV) at the surface (after etching during 150 s), provided a definite increase in oxygen concentration for longer exposure times (Figure 7c). Even at pressures of the order of 10^−10^ mbar, the surface of SmS gets oxidized with time. This process only occurs at the film surface, as all XRD measurements—essentially probing the entire thickness of the film—do not show any X-ray induced change from S-SmS to M-SmS.

Last but not least, Ar-ion sputtering during depth profiling analysis can also alter the obtained ratio of oxidation states [36]. Upon XPS depth profiling analysis, argon ions are used to sputter away material from the surface, thereby applying a local force and thus a possible conversion to the metallic state. Alternatively, sputtering can also change the precise stoichiometry at the exposed surface. This influence can be observed in Figure 7d as the color of an as-deposited 100 nm SmS layer changes from bluish-black to golden, as the sputter time increases from 0 to 200 s, indicative of a change from S-SmS (with dominantly Sm^2+^) to M-SmS (with Sm^3+^). Hence, it becomes clear that any reliable recording of the Sm valence state in SmS during XPS depth profiling experiments is rather complicated.

It seems that the strongly correlated character [37] and the volatile nature of the Sm oxidation state in the compound prevents a detailed analysis of the SmS stoichiometry and switching characteristics using conventional XPS. In-situ pressure dependent XPS analysis would be needed to shed light on the Sm valence state behavior, not only during the applied pressure but also upon pressure release. Similar studies have been performed by Deen et al., where they used X-ray absorption spectroscopy at the Sm L_III_ absorption edge, up to 2.86 GPa, studying the valence fluctuation, at 4.5 K [38]. That study pointed to an intermediate valence state of Sm in the magnetic state of SmS, at low temperatures. From our study, it is clear that the valence state of Sm is volatile, with a strong tendency of Sm^2+^ to switch to about 80% Sm^3+^, only due to being at the surface. Also, the sputtering process and measurement time play a drastic role in the final recorded ratio. According to previous XPS analysis, Mori Y. et al. [39] denoted that the oxidation in UHV of SmS (after annealing at 650 °C), initiates from the surface, with the high fraction of Sm^3+^ resulting from the volume expansion, due to the Sm oxide formation (within the SmS volume) and the pressure of expansion, leading to a decrease of the lattice constant. In our study, the SmS deposition took place at 1 × 10^−5^ mbar, with oxygen from trace gases able to reach the substrate. In addition, upon our deposition process, the heating of the target Sm metal with the e-beam could potentially lead to a reaction with trace gases, leading to the partial formation of Sm oxides, eventually leading to more oxygen being deposited in the thin film.

## 4. Conclusions

Through optical, structural, as well as electrical and chemical characterization, we optimized the deposition parameters for the reactive e-beam deposition of S-SmS thin films. The changes in properties of SmS thin films, upon the pressure-induced semiconductor-to-metal transition at room temperature, were presented. The return to the semiconducting state requires the use of thermal annealing, under vacuum (~400 °C) or other atmospheres, although the thermally induced transition already initiates at lower temperature of 250 °C. The repeatable optical and electrical changes upon transitioning between both states can find application in nanostructured strain-sensors, where the detection could take place optically or electrically. Homogeneous SmS deposition across 150 mm silicon wafers was achieved. In situ XRD experiments demonstrated that SmS layers can be stable up to 500 °C, in O_2_, with Sm_2_O_2_S emerging above that temperature. In a helium atmosphere, the in-situ XRD patterns showed the occurrence of Sm_3_S_4_, prior to the formation of Sm_2_O_2_S. Detailed XPS analysis was conducted to evaluate its applicability for determining the Sm valence states and oxygen content of the SmS layers. Sputtering by Ar-ions could already induce the valence state transition, while surface oxidation also played a role for longer measurement times.

This work not only shows the highly delicate fabrication process for obtaining high quality piezoresistive SmS thin films, but also demonstrates the limitations of XPS analysis in the determination of the SmS properties.

## Figures and Tables

**Figure 1 sensors-19-04390-f001:**
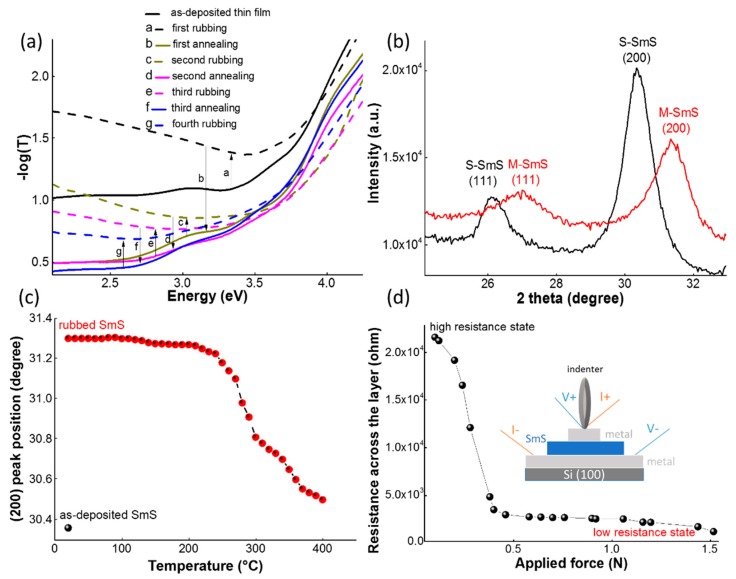
Optical, structural and electrical properties of samarium monosulfide (SmS) thin films in the semiconducting and metallic state. (**a**) Optical spectrum (T = transmission) of a 100 nm SmS thin film, which shows the repeated switching behavior between the semiconducting and metallic state. (**b**) XRD patterns of the semiconducting (black curve) and the pressure-induced metallic (red) state. (**c**) Position of the (200) diffraction peak, measured with the in situ high temperature X-ray diffraction (HTXRD) setup (10 °C per step), when thermally annealing in O_2_ a metallic SmS thin film. The black dot corresponds to the as-deposited semiconducting state SmS (S-SmS) thin film, for comparison. (**d**) Pressure-induced resistance change upon application of pressure by the indenter for a thin film stack of Al (50 nm)/SmS (100 nm)/Al (50 nm). The inset figure shows a sketch of the used measurement configuration.

**Figure 2 sensors-19-04390-f002:**
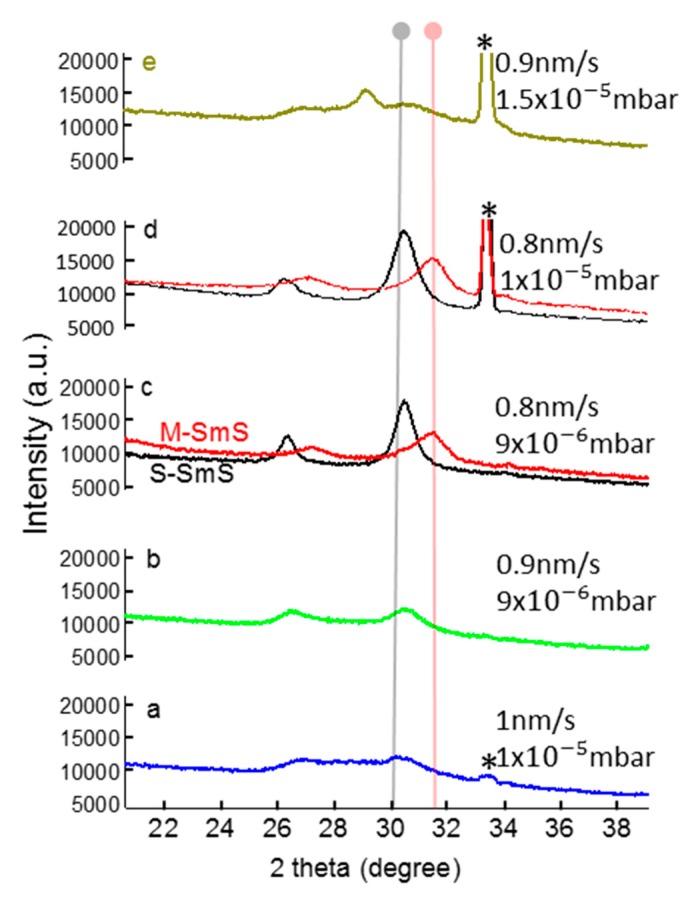
XRD patterns for 100 nm thick films as a function of the deposition rate (in nm/s) and the H_2_S pressure (mbar), for a substrate temperature of 250 °C. Grey and pink vertical lines represent the position of the (200) diffraction peak, for S-SmS and metallic state SmS (M-SmS) bulk crystals [14], respectively. For the thin film in graphs c and d, also the diffraction patterns for the thin film switched to the metallic state are shown, in red. The peak at ~33.3° is an artefact, related to the Si substrate (indicated with *).

**Figure 3 sensors-19-04390-f003:**
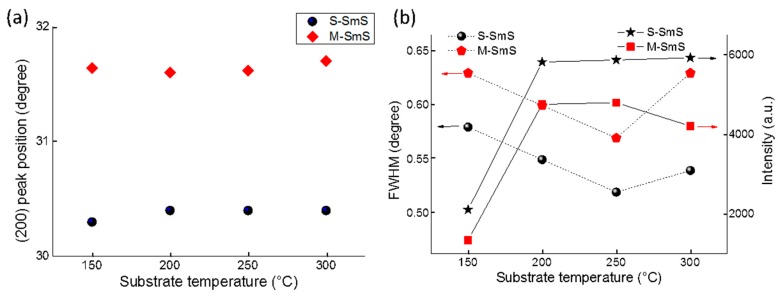
(**a**) Position of the (200) diffraction peak for the as-deposited and switched SmS thin films, as a function of substrate temperature. The thin films were deposited at a rate of 0.8 nm/s and at an H_2_S partial pressure of 10^−5^ mbar. (**b**) FWHM (in degree) and the diffraction peak intensity (in a.u.) for the same lattice plane. The background of each XRD measurement was subtracted before calculating the FWHM and peak intensity.

**Figure 4 sensors-19-04390-f004:**
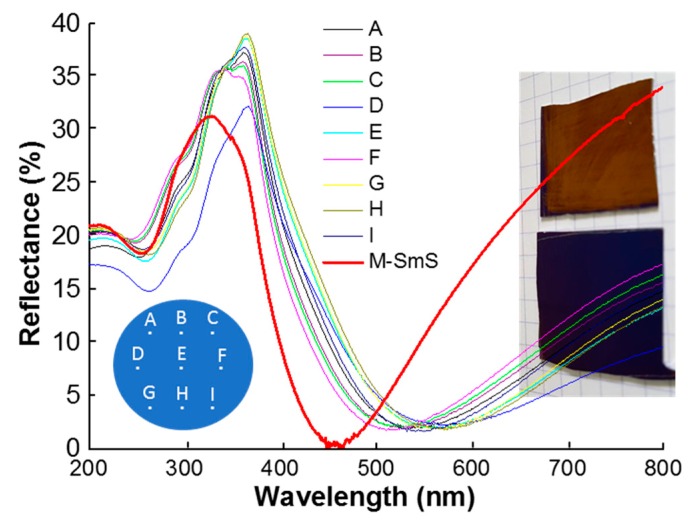
Optical reflectance from several spots on a 150 mm Si (100) wafer, on which 100 nm of SmS was deposited (substrate temperature of 250 °C). The scheme on the bottom left shows the location on the wafer for the respective reflection spectra. The photographs show the appearance of the as-deposited thin film (bottom) having a bluish-navy color and in the metallic state after rubbing (top), featuring a goldish color. The reflection spectrum of the latter (for position E), is indicated by “M-SmS”.

**Figure 5 sensors-19-04390-f005:**
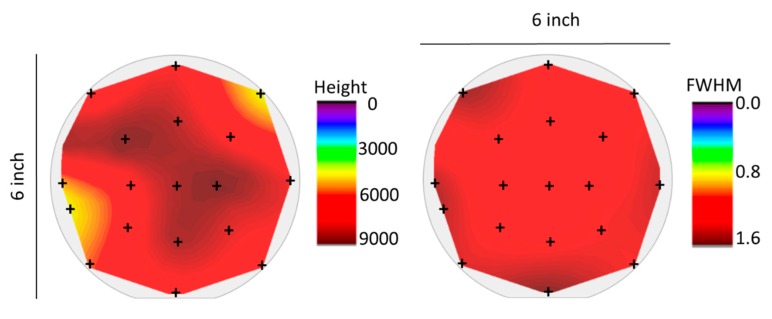
Left panel: Peak intensity mapping (in counts) of the (200) diffraction peak of S-SmS, across the 150 mm (6 inch) diameter wafer. The (200) and (111) peaks are the dominant ones. A thin film with a thickness of 100 nm was deposited at a substrate temperature of 250 °C. Right panel: FWHM mapping (in degree) of the same diffraction peak. Crosses correspond to the measurement positions on Si wafer. The light grey circle indicates the edges of Si wafer.

**Figure 6 sensors-19-04390-f006:**
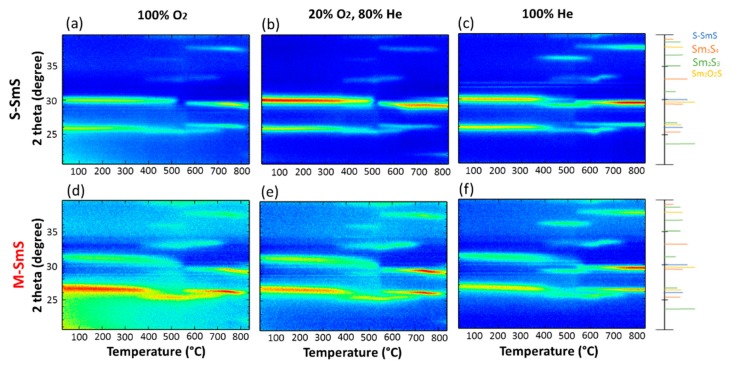
(**a**) In-situ XRD pattern of a 100 nm S-SmS on Si (100), under 100% O_2_. (**b**) Under 20% O_2_, 80% He. (**c**) Under 100% He. (**d–f**) Correspondingly for M-SmS. In all cases, the temperature ramp rate is 10 °C/s. The main peaks from the JCPDS (Joint Committee on Powder Diffraction Standards) files are indicated on the right.

**Figure 7 sensors-19-04390-f007:**
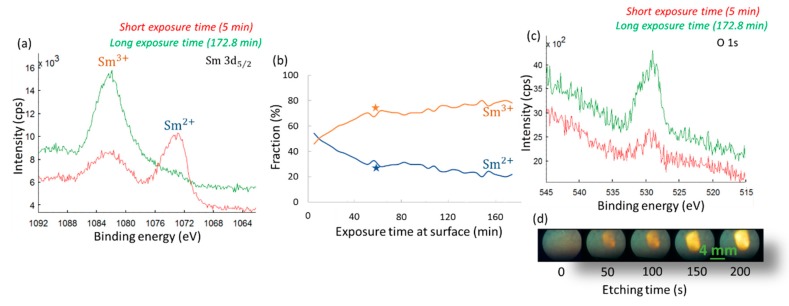
XPS analysis of a 100 nm thick SmS thin film. (**a**) Sm 3d_5/2_ photoelectron peak. The red curve corresponds to a short exposure time to X-rays of 5 min. Green curve corresponds to the longest measured exposure time of 172.8 min. (**b**) Fraction of Sm^2+^ and Sm^3+^ as a function of exposure time at the surface. X-rays remained on during the entire measurement. Orange and blue stars show the derived Sm^3+^ and Sm^2+^ percentages, after 1h surface exposure under vacuum conditions, with the X-rays off. (**c**) O 1s photoelectron peak. Same measurement conditions as in a). (**d**) Photographs taken by a microscope in the XPS set-up demonstrate the color change of the thin film surface, as the sputtering time increases from 0 to 200 s.

**Table 1 sensors-19-04390-t001:** Resistivity values of as-deposited and switched SmS thin films deposited at different substrate temperatures. The thin films were deposited at a rate of 0.8 nm/s and at an H_2_S pressure of 10^−5^ mbar. The values are the average of five different sheet resistance measurements of each sample, with the standard deviation between brackets.

Substrate Temperature (°C)	$\rho_{S-SmS}(Ω\mathrm{cm})$ (±s.d.)	$\rho_{M-SmS}(Ω\mathrm{cm})$ (±s.d.)
**150**	2.9 (±0.5) × 10^−1^	4.2 (±0.3) × 10^−3^
**200**	3.1 (±0.5) × 10^−1^	3.0 (±0.3) × 10^−3^
**250**	1.5 (±0.3) × 10^−1^	5.1 (±0.3) × 10^−3^
**300**	4.9 (±1.2) × 10^−1^	5.0 (±0.5) × 10^−3^

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
