# Peer review of "Switchable Piezoresistive SmS Thin Films on Large Area"

_sensors, 2019, doi:10.3390/s19204390_

Round 1
Reviewer 1 Report
The article is devoted to the synthesis of samarium monosulfide and its optimization. I like the form of the article. The most of papers available in the scientific space focuses on preparation and investigation the aimed, well optimized materials, while optimization process is very important for many readers. From this reason, I found the article as a valuable paper.
The SmS switchable piezoresistive materials is in the main of interest currently, from the reason of its great applicative potential. All this was clearly explained in the introductory section. The experimental section was presented in a correct way.
I recommend to publish the paper after some minor corrections, listed below.
In the first word of the abstract authors used abbreviation. Abstract should be understandable for a broad audience, so in my opinion authors should use full name: Samarium Monosulfide (SmS) The “pressure induced” expression should be written as “pressure-induced” In the introduction at the end of the sentence: “A detailed presentation of the properties of SmS can be found in our recent review paper” the reference should be provided. I understand, that authors try to avoid auto-citation, but in this place it is logical. Line 66: alpha is not displayed correctly
Author Response
Please see attachment. All changes to the manuscript (following the comments by both reviewers) have been highlighted in red.

Reviewer 2 Report
The manuscript entitled “Switchable piezoresistive SmS thin film on large area” is surely interesting, but several issues should be better addressed before further consideration in Sensors.
I will suggest the following major revisions:
Abstract:
Line 14: Please change “150 mm substrates” by “150 mm2”, or explain otherwise to clearly show the deposited area
Introduction:
Line 48 “our recent review paper”. Please include reference Line 66: Cu Kα instead of CuK֍1 Please, re – order this section. I suggest to put together the synthesis method and after that, introduce the characterization techniques. Specially, the two different paragraphs regarding XRD measurements should be put one after the other. Paragraph in line 69 should be elsewhere. I suggest after the XRD explanation. Line 95: Please introduce XPS acronym. Line 96: Please explain briefly how you use Ar+ to etch layers of the surface
Results and discussion:
Line 101 – 103. Please re – write these phrases, too informal writing, specially “one could start from …” Line 120: “The calculated lattice.. and Sm3+ ions”. How can you claim this by only XRD measurements? XRD does not provide a direct measurement of the valence state. You can support this statement by using other experimental techniques, but not exclusively by XRD. It it has been previously reported, please cite other works. Line 122: “Literature values ..570 pm”. Please include reference Line 124 – 125. This is a vague statement. Clamping by the substrate should be proved. Has this behavior been previously observed? Please provide references and discuss Please re – arrange the order of the images in Figure 1, so they are consistent with the text. First Fig 1a, 1b, and so on… Same with the figure caption: Line 146: “Structural, optical and electrical properties”, should be changed so they are in order with the images. To check the homogeneity of the deposited thin film, and see the coverage. I strongly suggest to perform microscopy images of the samples. Either SEM or AFM should be suitable Line 265: What is the spot size of the source? Furthermore, what is the footprint in the sample? 18 different spots mean what percentage of the sample proved? Line 335: I suggest to change the time scale. Please use minutes, not seconds. Specially, 10370 s…
Author Response

(The authors gave the same response as above.)

Round 2
Reviewer 2 Report
The manuscript can be published in the present form